# Conservative Management of Acute Sports-Related Concussions: A Narrative Review

**DOI:** 10.3390/healthcare12030289

**Published:** 2024-01-23

**Authors:** Sohaib Kureshi, Maria Mendizabal, John Francis, Hamid R. Djalilian

**Affiliations:** 1Neurosurgical Medical Clinic, San Diego, CA 92111, USA; 2TBI Virtual, San Diego, CA 92111, USA; 3Departments of Otolaryngology, Neurological Surgery, and Biomedical Engineering, University of California, Irvine, CA 92697, USA

**Keywords:** concussion, mild traumatic brain injury (mTBI), central sensitization, sports-related concussion, peripheral sensitization, glymphatic system, neuroinflammation, pain catastrophizing

## Abstract

This review explores the application of the conservative management model for pain to sports-related concussions (SRCs), framing concussions as a distinct form of pain syndrome with a pathophysiological foundation in central sensitization. Drawing parallels with proven pain management models, we underscore the significance of a proactive approach to concussion management. Recognizing concussions as a pain syndrome allows for the tailoring of interventions in alignment with conservative principles. This review first covers the epidemiology and controversies surrounding prolonged concussion recovery and persistent post-concussion symptoms (PPCS). Next, the pathophysiology of concussions is presented within the central sensitization framework, emphasizing the need for early intervention to mitigate the neuroplastic changes that lead to heightened pain sensitivity. Five components of the central sensitization process specific to concussion injuries are highlighted as targets for conservative interventions in the acute period: peripheral sensitization, cerebral metabolic dysfunction, neuroinflammation, glymphatic system dysfunction, and pain catastrophizing. These proactive interventions are emphasized as pivotal in accelerating concussion recovery and reducing the risk of prolonged symptoms and PPCS, in line with the philosophy of conservative management.

## 1. Introduction

### 1.1. Conservative Management Is Proactive in Nature

While conservative management paradigm starts with the Hippocratic principle of “first do no harm”, its mandate in sport medicine is much broader. Conservative management has evolved from mere avoidance strategies to ones that promote preventative interventions for sports-related injuries according to “the philosophy of active conservative care” [1]. In other words, conservative management in sports medicine is fundamentally proactive.

### 1.2. Conservative Principles in Pain Management

The conservative paradigm is applicable to pain management for sports injuries. While pain is an injury symptom in the acute period, it can become uncoupled in the sub-acute phase from the injury that first caused it, evolving into its own distinct disease state [2]. Classic examples of this in sports are when acute neck or back injuries transition into chronic low back pain (CLBP) or whiplash-associated disorders (WADs). The concept of “pain as a disease” is acknowledged in the 2017 International Olympic Committee consensus statement on the treatment of pain in elite athletes. The Committee notes that pain management should address “all contributors to pain including underlying pathophysiology, biomechanical abnormalities and psychosocial issues” [3]. This is in accordance with the biopsychosocial model of pain, a multidimensional approach used to understand and treat the complexities of chronic pain [4]. The consensus statement also emphasizes prevention as a primary goal of pain management, in line with the conservative care principles.

### 1.3. Central Sensitization and Conservative Management

Central sensitization describes the dynamics of chronic pain development, a process where amplification of the perception of pain is ascribed to alterations in central nervous system (CNS) sensory processing [5]. Woolf, one of the pioneers of the central sensitization model, explains:

“Central sensitization is where the CNS can change, distort, or amplify pain, increasing its degree, duration, and spatial extent in a manner that no longer directly reflects the specific qualities of peripheral noxious stimuli, but rather the particular functional states of circuits in the CNS… This does not mean that the pain is not real, just that it is not activated by noxious stimuli”.[6]

In other words, central sensitization is the process by which pain transitions from a symptom of injury to a disease state.

To further characterize this phenomenon, the IASP introduced the term nociplastic pain, a distinct category of pain that arises from altered pain perception despite “no clear evidence of actual or threatened tissue damage causing the activation of peripheral nociceptors or evidence for disease or lesion of the somatosensory system causing the pain” [7]. 

The central sensitization framework can be used to guide conservative interventions, and clinical researchers have used it to guide treatment in several areas of sports medicine, including chronic low back pain [8,9,10], shoulder pain [11], knee pain [12], chronic myofascial pain [13], tendinopathy [14,15], temporomandibular disorders [16], migraines [17], and post-traumatic headaches [18]. The key insight of the central sensitization model is that chronic pain syndromes can be prevented through early proactive interventions, using the biopsychosocial model of pain as a guide. 

Here, we argue that persistent post-concussion symptoms (PPCS) are a manifestation of central sensitization, and, because of this, early proactive interventions in accordance with conservative management principles should help reduce the incidence of PPCS. 

### 1.4. Applying the Conservate Care Paradigm in Concussion Injuries 

Concussions, or mild traumatic brain injuries (mTBIs), are a concerning problem for sports medicine. Approximately 3.8 million sports and recreational concussions are reported annually in the USA, with the number potentially being much larger since approximately 50% of concussions are not reported [19]. In 2022, the International Conference on Concussions in Sport—Amsterdam released its refined definition of concussion:

“Sport-related concussion is a traumatic brain injury caused by a direct blow to the head, neck or body resulting in an impulsive force being transmitted to the brain that occurs in sports and exercise-related activities. This initiates a neurotransmitter and metabolic cascade, with possible axonal injury, blood flow change, and inflammation affecting the brain. Symptoms and signs may present immediately, or evolve over minutes or hours, and commonly resolve within days, but may be prolonged”.[20]

Importantly, prolonged concussion symptoms are one of the primary complications of concussion injuries, and the prevention of prolonged symptoms should be seen as a primary goal of conservative management of concussions. 

The trend of current research is to recognize prolonged concussion symptoms as a form of nociplastic pain (i.e., pain secondary to central sensitization) [18,21,22,23]. PPCS follows the typical dynamic sequence of central sensitization: peripheral insult to peripheral sensitization, to prolonged neuroinflammation and neuroplasticity, to centrally-mediated chronic pain [24]. Recognizing PPCS as a nociplastic pain syndrome allows for the tailoring of conservative, biopsychosocial interventions that, when applied promptly after injury, may prevent its development. 

Multiple studies now demonstrate that early, proactive care significantly improves concussion outcomes (Table 1) [25]. This goal of this paper is to first outline the central sensitization model of pain chronification as it pertains to concussions, and to then explore how early, proactive interventions, targeting specific steps in the central sensitization process, can improve outcomes.

## 2. Persistent Post-Concussive Symptoms (PPCS) and Prolonged Recovery

### 2.1. Defining PPCS

Formerly referred to as “post-concussion syndrome”, the term PPCS was forwarded in the Berlin Consensus Statement on Concussion in Sport (2016), in part due to the stigma surrounding the term “syndrome” [31]. This was loosely defined as concussion symptoms lasting longer than 2 weeks in children and 4 weeks in adults, a definition later adopted by the American Medical Society for Sports Medicine (AMSSM) in 2019 [32]. In the latest Amsterdam consensus statement, the PPCS definition was updated to denote symptoms lasting for >4 weeks in all age groups. 

Adding to the confusion, the term “post-concussion syndrome” was removed from the fifth edition of the Diagnostic and Statistical Manual of Mental Disorders (DSM-5), referred to instead as “neurocognitive symptoms associated with traumatic brain injury”. Meanwhile, the 2024 International Classification of Diseases—Tenth Revision (ICD-10) diagnosis code F07.81 still describes the condition as “post concussional syndrome” [33]. The ICD-10 criteria is applied loosely in many studies, with the original descriptor, “Head injury usually sufficiently severe to result in loss of consciousness and then development within four weeks of at least three of the eight following symptoms: headache, dizziness, fatigue, irritability, sleep problems, concentration problems, memory disorders and emotion perturbations” [34,35,36]. The notation that concussions are “usually sufficiently severe to result in loss of consciousness” is largely ignored, as less than 10% of concussions result in a loss of consciousness [37].

### 2.2. The Epidemiology of PCSS

Epidemiological studies on the prevalence of PPCS are conflicting due to shifting definitions of the condition itself. A typical estimate of the prevalence of PPCS repeated in the literature is 15–30% [38], but this number can change dramatically depending on the criteria and timing used. For instance, one study showed that 64% of mTBIs were positive for PPCS at the three month mark by ICD-10 criteria vs. 11% when using the old DSM-IV criteria [34]. Furthermore, several large prospective studies now show that mTBI symptoms can persist for over a year in around 50% of cases [39,40].

### 2.3. The Epidemiology of Prolonged Symptoms

In sports medicine, the prevalence of prolonged single symptoms is arguably of more importance than the prevalence of PPCS. This is because return-to-play (RTP) criteria, by consensus (and in some cases by law), calls for the complete resolution of concussion-related symptoms [20]. Importantly, the relatively low PPCS prevalence reported in numerous publications may distort expectations on how long it should take for athletes to return to sports. 

For high-quality evidence on single prolonged symptoms, the Transforming Research and Clinical Knowledge in TBI (TRACK-TBI), a multi-center prospective cohort study, is an excellent resource. In this study, participants were evaluated at the 2-week mark, and then 3-, 6-, and 12-months post-injury. In this cohort of over 2000 patients, the percentage of participants who reported at least one new or worsened symptom after mTBI was 90% at 2 weeks, 78% at 3 months, 74% at 6 months, and 71% at 12 months. If the ICD-10 criteria of 3 or more symptoms was applied to this data set, 53% would be diagnostically positive at the one-year mark [41]. One problem with these numbers is that the TRACK-TBI cohort includes roughly 15% moderate to severe TBI cases, the 85% balance being mTBI cases.

Another source of data is the prospective, multicenter Predicting and Preventing Post-concussive Problems in Pediatrics (5P) cohort study of 3063 children 5–17 years of age, presenting within 48 h of a concussion injury. In this study, 50% of participants still had at least one symptom at the 28-day mark. Fatigue and headaches were the most common complaints. Notably, 68% of the concussions in this study were sports-related [42]. 

### 2.4. Controversies in the Prevalence of Prolonged Symptoms in Athletes vs. Non-Athletes

The above findings conflict with the clinical literature on collegiate sports-related concussions (SRCs), which reports a much faster recovery rate. For instance, in one cohort study of 1974 SRCs in college and club sport members, only 11.7% had symptoms lasting beyond 35 days [43]. Another source of data, the Concussion Assessment, Research, and Education (CARE) Consortium, is a 30-site study with 1751 collegiate athlete participants with SRCs. In this study, 80% of athletes were asymptomatic within 14 days of recovery, with a median time to recovery of 6.4 days and median RTP time of 12.8 days. Only 22.8% experienced “slow recovery”, which was defined in the study as symptoms lasting beyond 14 days or a RTP longer than 24 days [44]. 

There are several possible explanations for this conflicting data. When considering SRCs, is must be recognized that athletes form a unique population with distinct characteristics, such as the following: Age: Athletes tend to be younger than the general population.Health: The physical and mental health of athletes tends to be better than the general population.Access to care: Most athletes have access to athletic professionals who have basic training in concussion treatment.Supplements: Many athletes take supplements that may be neuroprotective, such as creatine.Severity of injury: SRCs tend to be less severe than injuries that involve polytrauma, such as in motor vehicle accidents.

On the other hand, non-sports concussions may take longer to recover than SRCs due to factors like age or mechanism of injury [45]. Further, litigation intent in non-SRCs may influence the rate of concussion symptom resolution, and studies have documented attenuated symptom improvement in those involved in litigation vs. controls [46]. 

In addition, SRCs in competitive sports are known to be subject to significant underreporting, with one study showing that 68% of college football players had at least one concussion they did not disclose [47]. The high rate of misreporting and underreporting in competitive-level sports most certainly distorts the data on symptom recovery time. We argue that this controversy deserves more attention in future research.

### 2.5. Risk Factors of Prolonged Recovery

Understanding risk factors for PPCS can alert practitioners to be particularly vigilant and proactive in the acute period for at-risk athletes. A variety of risk factors have been identified in the clinical literature, some of which are not always consistent. The most commonly reported preinjury risk factors are having prior concussions, female gender, mood disorders, learning disorders, attention deficit hyperactivity disorder (ADHD), and a personal or family history of migraines [48,49,50,51]. When looking at post-injury risk factors, the commonly cited factors include injury severity, retrograde amnesia, a high symptom score, “feeling in a fog”, delayed reporting, and the presence of sleep disturbances [40,47,52,53]. 

The issue of multiple concussions as a risk factor warrants further discussion. While several studies have documented that prior concussions are both a risk factor for future concussions and for prolonged recovery, others have failed to substantiate this connection [54]. While neuroprotective strategies for concussion (such as certain supplements like creatine or omega oils) are outside of the scope of this article, we generally recommend them for athletes after their first concussion. 

### 2.6. The Time-Course Pattern of Symptom Recovery

The time-course of mTBI symptom resolution underscores the importance of early intervention. There is a definitive “hockey-stick” pattern in the resolution of symptoms, with rapid resolution happening in the first weeks after concussion, followed by very little resolution after this period. 

In a prospective study of PPCS, it was found that being symptomatic at one month was significantly predictive of being symptomatic at one year [55]. This is in line with the findings of the TRACK-TBI study, where there was only a 7% drop in those with one symptom from the 3-month to 12-month mark. Expressed another way, symptoms decreased steadily at ~1.7 symptom score points per month in the first 3 months but then only 0.2 points per month for the rest of the year [41]. 

This distinctive time-course pattern correlates with the development of nociplastic pain and the phenomenon of central sensitization. When symptoms linger, central neuroplastic changes occur, and the symptoms become chronic. This underscores the notion that a therapeutic window for conservative interventions exists in the early period, but this closes once the deleterious neuroplastic changes in central sensitization become established.

## 3. Central Sensitization and Symptom Chronification in Concussions

### 3.1. The Neurobiological Sequence of Central Sensitization 

In the central sensitization, the neurobiological sequence begins when an insult to peripheral tissues triggers “peripheral sensitization”, a state marked by the heightened responsiveness and excitability of nociceptors, an increased signaling rate, and strengthened synaptic transmission. When this is sustained, glial cells, activated by calcitonin gene-related peptide (CGRP) and other neuropeptides, propagate a neuroinflammatory response, enhancing sensitization in higher order neurons and initiating neuroplasticity [56]. Descending facilitation from the brainstem amplifies pain signals, and cortical plasticity in higher brain centers contributes to heightened and prolonged pain perception. Stress, anxiety, and pain catastrophizing intensify and stabilize neural pain signaling pathways through the neuroendocrine system [57] and the “nociceptive amygdala” [58]. The cumulative result is CNS remodeling (neuroplasticity), leading to pathologically elevated pain and sensory hypersensitivity, independent of the original source of peripheral injury. 

### 3.2. Central Sensitization and Chronic Pain 

Importantly, when central sensitization has occurred, susceptibility to other chronic pain conditions is elevated. Animal mTBI models clearly demonstrate that the neurogenic inflammation following traumatic brain injury is coincident with measurable indicators of chronic pain, both histologically and behaviorally, consistent with the central sensitization model of pain chronification [59]. This phenomenon is seen in humans with concussion injuries, where more than 50% of people report developing issues with chronic pain [60], and there is a nearly 5-time increased risk of experiencing persistent pain [61]. Again, litigation intent is an important factor to remember in this context. 

The clinical hallmarks of central sensitization are noted in mTBI patients [62], including heightened sensitivity to painful stimuli (hyperalgesia), pain from usually nonpainful stimuli (allodynia), and increased sensitivity to both external and internal stimuli (global sensory hyperresponsiveness) [63]. Importantly, chronic central sensitization-related pain is also associated with an increased risk of opioid abuse [64]. Along these lines, clinical studies show that the odds of prescription opioid abuse are 1.5–2.9 times higher in previously concussed adolescents [65,66].

### 3.3. Central Sensitivity Syndromes

Central sensitization can become syndromic, extending beyond the perception of pain. People with central sensitization syndromes often present with multiple overlapping symptoms, including physical and cognitive fatigue, mood disorders, dysautonomia, sensory hypersensitivity, and sleep disorder. Terms like Central Sensitivity Syndrome (CSS) and Chronic Overlapping Pain Conditions (COPCs) have been introduced to describe this [67]. Importantly, PPCS follows this syndromic pattern, with typical symptoms extending beyond physical pain. 

## 4. Potential Targets for Conservative Interventions for Acute Concussions

By viewing PPCS as a form of nociplastic pain secondary to central sensitization, it reveals a time-dependent process that can potentially be interrupted. Recognizing this temporal dimension is key, as it points to a critical therapeutic window for early intervention. The strategic goal is to prevent the neuroplastic changes that are responsible for pain chronification, in line with the International Olympic Committee consensus statement referred to above [3]. 

Here, we cover five components of the central sensitization process specific to concussions that can be targeted with conservative treatments in the acute period:Peripheral sensitization;Cerebral metabolic dysfunction;Neuroinflammation;Glymphatic system dysfunction;Pain catastrophizing.

As will become clear, certain interventions can potentially positively affect multiple central sensitization-related targets. This supports a clinical intervention strategy that is multimodal in nature, consistent with conservative management principles. In Table 2, selected interventions that address two or more of the five targets described above are presented. 

### 4.1. Reducing Peripheral Sensitization

Sustained peripheral pain signaling and peripheral sensitization are the initiating mechanisms of the central sensitization process. Animal models of central sensitization highlight the need for ongoing nociceptive input to drive this phenomenon, suggesting a therapeutic benefit in aggressively reducing such input in the acute period. This approach, particularly in perioperative medicine, has shown promise in mitigating central sensitization-related chronic pain [67]. 

Since the peripheral source of pain in concussion occurs primarily through the ophthalmic branches of the trigeminal nerve (V1) that innervate the meninges and cranial periosteum [77] and secondarily through CGRP-mediated activation of the trigeminal ganglion and trigeminocervical nucleus, the treatment of peripheral trigeminal hypersensitivity may help abort the development of PPCS [78,79,80].

#### 4.1.1. Exercise 

Clinical data on early exercise in the past decade have revolutionized the treatment of concussion injuries, with high quality evidence supporting this intervention [68,81]. For most studies, early exercise describes the initiation of physical activity in the first 48–72 h after injury. One of the many benefits of exercise is that it reduces peripheral sensitization through mechanisms such as endorphin release and changes in pain modulation systems [82]. Importantly, symptom-limited exercise should be prescribed, as overexertion may lead to symptom exacerbation and the worsening of peripheral sensitization [83].

#### 4.1.2. Analgesics

Analgesic therapy can be effective at decreasing peripheral sensitization. Non-steroidal anti-inflammatory drugs (NSAIDs) normalize the heightened pain threshold linked to inflammation by inhibiting prostaglandin formation at peripheral and central sites [84]. Acetaminophen achieves peripheral prostaglandin inhibition and can positively affect several central anti-nociception processes [85]. The perioperative use of preventative analgesia with these and other drugs successfully reduces central sensitization-related chronic pain after surgery [86,87]. 

In the concussion literature, a randomized clinical trial found that a combination of NSAIDs and acetaminophen showed the most promising results in terms of reducing the number of headache days and return to school time [88]. On the other hand, the 5P cohort study (referenced above) showed that emergency room administration of the OTC oral analgesics had no impact on the presence of headaches at the 7-day follow up. There was no mention of whether it was a one-time dose or if treatment was continued at home [89]. 

#### 4.1.3. Cold Therapy

Cold therapy to the head and neck may potentially reduce peripheral sensitization. Cold compresses alleviate pain by slowing pain signal transmission to the central nervous system and cell metabolism. Additionally, cold reduces inflammation, constricts vessels, and decreases the release of chemical pain mediators, leading to an increased pain threshold and reduced pain [90]. 

In concussions, branches of the trigeminal nerve that innervate the cranial periosteum show an acute inflammatory response after closed head trauma [78], suggesting that topical cooling may mediate acute trigeminal hypersensitivity in the cutaneous nerves of the scalp and neck. In the clinical literature, one study found that concussed subjects self-reported temporary relief from physical symptoms after head cooling [91]. In another group of studies, players receiving head–neck cooling have shorter return-to-play times than controls in several studies [71,92].

#### 4.1.4. Physical Therapy

Physical therapy can help reduce hypersensitivity of the balance system. Vestibular therapy has shown promising results when instituted early in cases where there are balance issues. In vertiginous states, there is hypersensitivity in the balance organs, just as there is with other sensory inputs. Just like with exercise, the goal of vestibular therapy is to engage in symptom-limited movements to challenge and reset hypersensitivity in the balance system. Cervical therapy has also been shown to be beneficial in select cases when started early. Importantly, the neck has cross-signaling from trigeminal inputs in the cervicotrigeminal nucleus. 

#### 4.1.5. Sensory Protection

Photophobia is common in the acute period after concussion [93], reflecting underlying trigeminal sensory nerve hypersensitivity. Because intrinsically photosensitive retinal ganglion cells (ipRGCs) are sensitive to 480 nm light, FL-41 glasses and other tinted lenses are applied in cases of post-concussive photophobia [94,95]. The use of ear plugs in phonophobia may be helpful in the short term, but there are no data to support this. For sensory hypersensitivity, treatment involves gradual and systematic sound desensitization rather than total deprivation [96]. 

### 4.2. Addressing Cerebral Metabolic Dysfunction

The neurometabolic consequences of concussion drive central sensitization-related changes by amplifying peripheral sensitization and secondary injury in the brain. Numerous ionic, metabolic, and physiological changes occur acutely after concussions, which contribute to migraine-phenotype symptoms [97]. Cortical spreading depolarization, like those seen in migraines, may represent the initiating event in this cascading metabolic dysfunction [98]. Depolarization results in excess glutamate, which in turn triggers an ionic imbalance that leads to the overactivity of sodium–potassium pumps, causing an increase in energy demand in the form of adenosine triphosphate (ATP). This intensifies the need for glucose metabolism and oxygen at a time when cerebral blood flow (CBF) is impaired [24]. These changes overload mitochondria, altering their membrane permeability and triggering oxidative stress through the production of reactive oxygen species (ROS) [97,99]. This neurometabolic cascade coincides with the first week after injury, the window of time when clinical symptoms are the most severe after concussions and when central sensitization-related changes begin to occur. 

#### 4.2.1. Mitochondrial Support

Interventions for acute mitochondrial dysfunction in concussions are a promising and exciting area of study. Because early mitochondrial dysfunction is seen in acute mTBI [100], addressing mitochondrial impairment may enhance mTBI outcomes [101]. 

Creatine supplementation may help with the metabolic crisis after concussions, with preclinical evidence supporting both preventative effects of symptoms when taken before injury and recovery effects when taken after injury [102]. In addition to its role in sustaining ATP concentrations and cellular bioenergetics [103], creatine is believed to contribute to preserving mitochondrial membrane potential and reducing intramitochondrial reactive oxygen species and calcium [104]. There are only limited human trial data on creatine in TBI patients, but the results were uniformly positive [105,106]. The prevalence of creatine supplementation in athletes is unknown. It is not a banned supplement, but high-school athletic professionals are prohibited from recommending it in certain states. As more data on the benefits of creatine supplementation for concussion treatment (both before and after injury) become clear, this prohibition may be revised. 

Ketogenic diets are also being explored for their ability to mitigate mTBI-related glucose hypometabolism [107], with pilot clinical trials in PPCS patients showing promising results [107,108]. Like creatine, ketone bodies serve as an energy substrate, bypassing glycolysis to undergo direct metabolism via the tricarboxylic acid cycle. This process enhances oxygen metabolism, supports mitochondrial function, and reduces oxidative stress and glutamate-induced injury [109]. 

Another compound of interest is ubiquinol (coenzyme Q-10), which has been shown to preserve mitochondria and reduce oxidative stress in animal TBI models [110].

Vitamin D is now known to be a critical mitochondrial transcription factor, and, in preclinical models of neurodegenerative disease, it rescues mitochondria from oxidative stress [111,112]. 

#### 4.2.2. Exercise

Exercise is a powerful intervention for the post-concussion metabolic crisis because it increases CBF, which in turn increases oxygen and glucose delivery [113]. Multiple studies have documented that CBF is decreased in the acute and subacute periods after SRCs [114]. While not quantified in any studies, the beneficial effects seen with early post-concussion exercise may be related to its ability to increase CBF. 

#### 4.2.3. Deep Breathing

Clinical evidence shows that regular deep breathing exercises significantly increase blood oxygen levels [115]. Furthermore, breathing exercises have been shown to decrease mitochondrial-related biomarkers of oxidative stress in a variety of clinical contexts [116].

#### 4.2.4. Cold Therapy

Vigorous exercise may lead to hyperthermia, which can exacerbate concussion-related glucose hypometabolism. Therefore, cooling after exercise may improve outcomes in this regard by reducing hyperthermia-related neurometabolic burden [117].

### 4.3. Decreasing Neuroinflammation 

Neuroinflammation is a pivotal component in the initiation of nociplastic pain after concussions because inflammatory mediators and cytokines drive the sensitization of neurons within the CNS [118]. In traumatic brain injuries, CGRP is an integral player in this process by amplifying the microglial response and priming neurons for neuroplastic changes in the trigeminal system [119]. CGRP levels increase after concussions due to an increased expression in trigeminal nociceptors and in response to post-injury cortical spreading depolarization [120]. In animal models, CGRP inhibitors prevent the development of PPCS-like hypersensitivity when delivered in the first two weeks after injury, but not after this period [22]. Clinically, there is evidence that CGRP polymorphism can partially explain clinical outcomes after concussions [121]. Another study found that an intravenous infusion of CGRP triggers migraine-like headaches in people with PPCS, highlighting the significant role of CGRP in the genesis of post-traumatic headaches [122]. These connections suggest that targeting neuroinflammation early after concussion injuries may disrupt the cascade of events, potentially preventing PPCS development [123].

#### 4.3.1. Nutraceuticals

Several nutraceuticals may be beneficial for reducing concussion-induced neuroinflammation. For instance, Vitamin D may reduce neuroinflammatory and secondary injury effects after concussions [124]. Preclinical studies demonstrate that Vitamin D improves post-mTBI neuroinflammation and oxidative stress [125,126]. Vitamin D has also been shown to decrease CGRP levels in migraineurs [127]. Importantly, TBI patients’ low Vitamin D levels are noted to have significantly worse cognitive impairment and poor functional outcomes [76,128], whereas early vitamin D supplementation in deficient patients leads to significant improvement in clinical outcomes [129]. Ubiquinol (coenzyme Q-10) has been shown to decrease oxidative stress and neurodegeneration in TBI animal models [130]. It also has been shown to decrease CGRP and neuroinflammatory markers in migraineurs [131,132]. Other nutraceuticals that have been shown to significantly decrease CGRP in human trials for migraines include melatonin [133] and curcumin (turmeric) [134]. Preclinical data show that omega oils prevent the neuroinflammatory changes in microglia to a pro-inflammatory phenotype, activate neuroprotective cytokines, and mitigate TBI-related blood–brain barrier disruption [135,136].

#### 4.3.2. Dietary Changes

There is an increasing interest in the impact of diet on concussion outcomes [108,137,138]. Diets rich in red meat, saturated and trans fats, refined sugars, and carbohydrates are associated with neuroinflammation, while neuroprotective and anti-inflammatory effects are linked to diets high in unsaturated, polyunsaturated, and monounsaturated fats, as well as ketogenic and Mediterranean diets, and intermittent fasting [139]. The effects on anti-neuroinflammatory (ANI) diets on TBI outcomes are being worked out in both preclinical [140] and clinical studies [108,141]. Specific supplements have been extensive studied, most notably the use of omega oils [142,143,144]. There is also interest in the impact of TBI on the gut microbiome [145]. Other recommendations include a low glutamate, low tyramine, and low histamine diet combined with the elimination of caffeine from the diet. 

#### 4.3.3. Exercise

Regular moderate-intensity exercise is associated with anti-inflammatory effects [146]. Animal models of TBI support that early exercise reduces markers of neuroinflammation and nociceptive sensitization [147,148]. Further research is being proposed on the role of exercise immunology in TBI outcomes [149].

#### 4.3.4. Stress Reduction

There is a direct association between psychological stress and neuroinflammation [150]. Stress-reducing interventions like deep breathing and mindfulness-based stress reduction (MBSR) have been shown to decrease biomarkers of neuroinflammation in clinical trials of central sensitization-related symptoms [151]. Multiple clinical trials suggest that mindfulness-based interventions are supportive of mTBI recovery [152].

### 4.4. Optimizing Glymphatic System Functioning

The prolonged presence of neuroinflammation in concussions may be associated with dysfunction in the glymphatic system during the acute phase following the injury [153]. The glymphatic system, a waste clearance pathway in the central nervous system, plays a crucial role in removing metabolic byproducts and cellular debris after concussion [154]. Damage-associated molecular patterns (DAMPs) are due to normally intracellular proteins that, when released into the extracellular space after trauma, elicit an immune response. The release of DAMPs is linked to adverse post-TBI outcomes, including diminished memory, altered motor coordination, and cognitive impairments [99]. This debris is normally cleared by the glymphatic system, but, in the aftermath of a concussion, the glymphatic system functioning is decreased by up to 60% [155]. This compromised waste removal process contributes to the sustained inflammatory response to neurotoxic proteins (like tau), which, in turn, is permissive of central sensitization-related neuroplasticity [156]. This is concerning, as cumulative tau deposition around intracerebral vessels is the histological definition of chronic traumatic encephalopathy (CTE) [157]. Therefore, optimizing glymphatic function in the acute period after concussions is an important therapeutic goal [18].

#### 4.4.1. Circadian Therapy

The glymphatic system is linked to sleep and the circadian system, with over 80% of glymphatic system clearance occurring during deep sleep [158,159,160]. Because sleep is also pathologically disturbed after concussion injuries [161,162], circadian therapy interventions are important early interventions. Melatonin has been extensively studied as an adjuvant therapy for concussions, with a meta-analysis showing positive sleep-related outcomes in 8 of 9 studies [74]. Limiting blue light via screen-time restriction positively impacts concussion recovery time [163]. Morning blue light therapy is associated with improvement in multiple concussion outcomes in patients with established PPCS [164]. Finally, because sleep apnea significantly disrupts glymphatic function [165] and is associated with poor outcomes in concussion [166], screening for and treatment of sleep apnea is recommended in concussion injuries. Other interventions along these lines include sleep hygiene, cognitive behavioral therapy (CBT), and prescribed exercise [18].

#### 4.4.2. Omega Oils

Numerous studies demonstrate the positive impact of omega oils on sleep quality and their influence on circadian variations in blood pressure, potentially through direct effects on melatonin release and norepinephrine regulation [167,168]. Additionally, in a TBI animal model, omega oils were found to enhance glymphatic drainage, reducing neurological impairment after simulated injury; this effect was directly related to improved glymphatic clearance [136].

#### 4.4.3. Exercise and Deep Breathing

Because exercise increases blood pressure and CSF flow, it has a positive effect on glymphatic functioning [169,170]. Similarly, deep breathing influences glymphatic clearance by increasing the flow magnitude of CSF [171].

### 4.5. Pain Catastrophizing 

Another possible target to prevent the central sensitization process after concussion is pain catastrophizing in the acute period [172]. Pain catastrophizing is the putative link between trait anxiety and post-concussion symptoms [173]. Pain catastrophizing involves an exaggerated negative mental outlook toward pain, magnifying its threat, fostering feelings of helplessness, and contributing to increased distress and disability. Several orthopedic studies have linked pain catastrophizing with the development of nociplastic pain and central sensitization [174,175,176]. 

In concussion research, pain catastrophizing subscales (rumination, magnification, and helplessness) show significant correlations with pain severity, the number of reported post-concussion symptoms, psychological distress, and functional levels [177]. These findings are related to studies on the relationship between trait resilience (arguably the opposite of pain catastrophizing) and concussion outcomes. Reduced resilience is linked to increased symptoms and a prolonged recovery from SRCs, whereas high resilience shows the opposite [178,179].

#### 4.5.1. Mindfulness, Meditation, and Deep Breathing

There is significant clinical evidence that trait mindfulness is negatively associated with pain catastrophizing [180,181] and that mindfulness exercises may help abort pain catastrophizing [151,182,183]. A recent meta-analysis examining the impact of post-mTBI interventions like mindfulness, meditation, and yoga revealed substantial improvement in overall symptoms when compared to control groups, including improvements in mental health, physical well-being, cognitive performance, and overall quality of life [72]. In a pilot study, deep breathing exercises in concussion patients was associated with decreased stress and tension [70].

#### 4.5.2. Coaching

The use of coaching in pain management is gaining interest. Clinical trials of health coaching to aid with stress management and goal-setting in chronic pain patients show reduced psychological stress and improved resilience [184]. In the concussion literature, web-based resources combined with weekly coaching sessions produced improved outcomes and high patient satisfaction rates [185]. In a randomized trial, collaborative care, where coping skills, relaxation strategies, sleep hygiene, and positive thinking techniques were coached, showed significantly improved results in PPCS patients [186]. In another randomized trial, the addition of weekly motivational interviewing and CBT significantly improved multiple mTBI outcome measures [187]. Another randomized trial showed that regular telephonic follow-up using motivational and behavioral activation approaches resulted in significantly lower depression scores, including in those with preexisting depression [188].

#### 4.5.3. Exercise

There is evidence that exercise can improve pain catastrophizing by influencing the fear center of the brain. Voluntary exercise stimulates neurons in the mesolimbic system, including in the amygdala. This has been suggested as the mechanism by which exercise aids in overcoming fear-avoidance behaviors associated with chronic pain conditions [189].

## 5. Conclusions

Sports-related concussions are of increasing concern to the athletic and medical communities. Athletes form a unique class of population in terms of demographics, and the mechanism of injury of SRCs tends to be milder than is seen in polytrauma contexts (like motor vehicle accidents). Because athletes have access to basic care by athletic professionals, there is the potential to introduce proactive protocols that can prevent the onset of prolonged concussion symptoms. What is lacking is a framework that integrates these interventions into a logical treatment plan. We argue that framing PPCS as a form of central sensitization accomplishes this.

In outlining PPCS as a form of chronic pain, and by examining the mechanisms for how chronic pain develops using the central sensitization model, several interventions for the prevention of prolonged symptoms and PPCS emerge. These include methods designed to reduce peripheral sensitization, address cerebral metabolic dysfunction, mitigate post-concussion neuroinflammation, optimize glymphatic system functioning, and reduce pain catastrophizing. This proactive program is in line with the principles of conservative management in sports medicine.

There are several inherent limitations in this current review. First, any review of this type is limited by the author’s awareness of all potentially relevant research. In addition, apart from exercise, there is a shortage of high-quality evidence-based research supporting many of the potentially promising interventions mentioned above.

Despite these limitations, theory-informed treatments and insights from preclinical studies, particularly those grounded in the central sensitization model, can still play a pivotal role in developing a program for proactive conservative care. Our hope is that this review contributes meaningfully to addressing empirical and knowledge gaps in the concussion literature and guiding strategic interventions in concussion management.

## Figures and Tables

**Table 1 healthcare-12-00289-t001:** Studies documenting the impact of early intervention for concussion outcomes.

Author	Study Design *	Summary of Findings
Bock, et al. (2015) [26]	RS, *n* = 366	Treatment in <7 days results in significantly shorter recovery time (*p* < 0.05).
Cassimatis, et al. (2021) [27]	RS, *n* = 341	Late treatment (>28 days) results in 3× longer recovery time compared to early treatment (<14 days) (148 vs. 39 days, 95% CI: 30.7–46.7).
Eagle, et al. (2020) [28]	RS, *n* = 218	Prolonged recovery is 10× greater (OR = 9.8) when seen 8–20 days after recovery vs. <7 days.
Kontos, et al. (2020) [29]	RS, *n* = 162	An early treatment (<7 days) group recovered 20 days sooner than those seen late (8–20 days).
Pratile, et al. (2022) [30]	CS, *n* = 1213	Treatment in <10 days recovered in 23.5 days vs. 37.1 days for those assessed in 10–30 days.

* RS = retrospective study, CS = cohort study.

**Table 2 healthcare-12-00289-t002:** Selected interventions that address multiple CS-related targets.

Intervention	Target *	Author, Study Details **	Summary of Findings
Early Exercise	PS, CN, NI, GO, PC	Leddy, et al. (2023) [68]; MA (*n* = 9432)	Early physical activity and prescribed exercise improved recovery by a mean of −4.64 days (95% CI −6.69, –2.59).
		Grool, et al. (2016) [69]; CS (*n* = 2413)	Early participation (<7 days) in physical activity compared with no physical activity was associated with lower risk of PPCS (413 [24.6%] patients vs. 320 [43.5%] patients; RR, 0.75 [95% CI, 0.70–0.80]).
Deep Breathing	CN, NI, GO, PC	Cook et al. (2021) [70]; P (*n* = 15)	Following deep breathing exercises, participants reported significant reduction in stress (r = 0.57), tension (r = 0.73), fatigue (r = 0.73), and confusion (r = 0.67), with large effect sizes.
Cold Therapy	PS, CN	Al-Husseini, et al. (2022) [71], RCT (*n* = 132)	The proportion of players with prolonged symptoms (>14 days) was 24.7% in the cold therapy intervention group and 43.7% in controls (*p* < 0.05)
Mindfulness	NI, PC	Acabchuk, et al. (2021) [72], MA (*n* = 532)	Meditation, yoga, and mindfulness-based interventions lead to significant improvement of overall symptoms compared to controls (d = 0.41; 95% CI [0.04, 0.77]; τ2 = 0.06).
Melatonin	NI, GO	Barlow, et al. (2019) [73], MA (*n* = 15)	Meta-analysis of pre-clinical data showed a positive effect of melatonin on neurobehavioral outcome (SMD = 1.51 (95% CI: 1.06–1.96)), neurological status (SMD = 1.35 (95% CI: 0.83–1.88)), and cognition (SMD = 1.16 (95% CI: 0.4–1.92)) after TBI.
		Cassimatis, et al. (2022) [74], MA (*n* = 251)	Eight of nine mTBI studies reported positive sleep outcomes after melatonin treatment, with significant improvements in subjective sleep quality, objective sleep efficiency, and total sleep, and reductions in self-reported fatigue, anxiety, and depressive symptoms.
Omega oils	NI, GO	Miller, et al. (2022) [75], RCT (*n* = 40)	In SRCs, the treatment group took 2 g of docosahexaenoic acid (DHA) daily for 12 weeks. The DHA group were symptom-free earlier than the placebo group (11.0 vs. 16.0 days, *p* = 0.08) and had a shorter RTP time (14.0 vs. 19.5 days, *p* = 0.12).
Vitamin D	CN, NI	Sharma, et al. (2020) [76], RCT (*n* = 35)	In moderate to severe TBI, Vitamin D bolus in the acute period showed significant improvements in cognitive and physiological outcomes. Inflammatory markers were also significantly decreased in the treatment group (IL-6 *p* = 0.08, TNF-α *p* = 0.02).

* PS = peripheral sensitization, CN = cerebral neurometabolism, NI = neuroinflammation, GO = glymphatic optimization, PC = pain catastrophizing. ** CS = cohort study, MA = meta-analysis, RCT = randomized clinical trial, P = pilot study.

## Data Availability

All data are available online.

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
