# Peer review of "Conservative Management of Acute Sports-Related Concussions: A Narrative Review"

_healthcare, 2024, doi:10.3390/healthcare12030289_

Round 1
Reviewer 1 Report
Comments and Suggestions for Authors
Brief Summary
The review explores sports-related concussions as a distinct form of the central sensitization pain model and argues for a conservative management model to be applied to sports-related injuries for proactive care. The authors provide evidence that supports an early therapeutic window as a critical time to address concussion symptoms based on the central sensitization framework. The pathophysiology of concussion is systematically discussed, and proactive treatment approaches are provided for many neurobiological systems affected by concussion.
General Comments
The article has a clear emphasis on proactive treatments and prevent compelling data to support this early therapeutic window. The review could be improved by addressing a few general concerns:
1. Although the review is well organized in succinct sections, your central argument to apply pain central sensitization model to sports-related concussions is lost at certain parts. Below I have made specific notes of certain paragraphs that detract from your central argument and can be either condensed or removed.
2. The review also warrants more references to preclinical, animal models. Later in the paper the authors begin to include sources from animal models, but more sources should be added into the discussion early. The use of preclinical evidence eliminates many of the confounds that the authors raise as problematic in clinical studies.
3. Section 2 “Persistent Post-Concussive Symptoms (PPCS) and Prolonged Recovery” and the Conclusion lack a sports-related focus. The introduction nicely provides context to traumatic brain injuries occurring specifically in athletes. The rest of the paper should carry this focus by specifically discussing the access that athletes have to these early interventions. How do athletes provide a unique population (diet, exercise, mental health) prior to injury? They are at the advantage of generally already having good fitness, diet, supplement, mental health and access to trainers etc. Are treatment strategies for sports-concussions being applied in a systemic way anywhere as a case study?
4. Add a discussion about repeated concussions which is commonly seen in sports-related mTBI cases. Can early prevention strategies help in the case of repeated injuries? If PPCs are not properly diagnosed or left untreated, are consequences exacerbated if an athlete sustains another injury?
Specific Comments
38: Can you provide an example?
44: justify the use of “so-called”; are you referring to limitations of the model?
51: Lacking a clear definition of central sensitization in chronic pain. Can you expand (take from Section 3.1)?
59: Make a link here to concussions. “We argue that mTBI should also be examined under the conservative care paradigm because…”
Table 1: Too much information in the summary of findings. Could you add a column for timeline and pull out this information?
199-212: Very similar to Section 1.3. Only need to include once.
229-239: This would be an appropriate place to include more preclinical evidence. Studies in rodents also support these findings (where litigation intent doesn’t matter).
243-253: This paragraph can be condensed- doesn’t contribute much to your argument.
Table 2: Confusing to have two “PS” abbreviations. Can details about timeline be included? Need more of an explanation for which categories are included in the table. Why not include them all? Alternatively, you could restructure the paragraph and list all variables not included in the table at the end.
282: define early
353: Creatine specifically is a supplement athletes may have already been taking prior to injury. Any evidence about how common this is or if it has any preventative effects if taken prior to concussion.
Author Response
Dear Reviewer,
Thank you for your kind remarks and thorough review. Please see below our changes in response to your input.
Sincerely,
John Francis
_____________________________________________________________________
Report 1:
Brief Summary
The review explores sports-related concussions as a distinct form of the central sensitization pain model and argues for a conservative management model to be applied to sports-related injuries for proactive care. The authors provide evidence that supports an early therapeutic window as a critical time to address concussion symptoms based on the central sensitization framework. The pathophysiology of concussion is systematically discussed, and proactive treatment approaches are provided for many neurobiological systems affected by concussion.
General Comments
The article has a clear emphasis on proactive treatments and prevent compelling data to support this early therapeutic window. The review could be improved by addressing a few general concerns:
- Although the review is well organized in succinct sections, your central argument to apply pain central sensitization model to sports-related concussions is lost at certain parts. Below I have made specific notes of certain paragraphs that detract from your central argument and can be either condensed or removed.
Author reply:This concern was acknowleged and per your specific guidance below, was adjusted, condensed, or removed.
- The review also warrants more references to preclinical, animal models. Later in the paper the authors begin to include sources from animal models, but more sources should be added into the discussion early. The use of preclinical evidence eliminates many of the confounds that the authors raise as problematic in clinical studies.
Author reply:Reference to preclinical studies, specifically animal models, was inserted into the discussion per your specific suggested points below.
- Section 2 “Persistent Post-Concussive Symptoms (PPCS) and Prolonged Recovery” and the Conclusion lack a sports-related focus. The introduction nicely provides context to traumatic brain injuries occurring specifically in athletes. The rest of the paper should carry this focus by specifically discussing the access that athletes have to these early interventions. How do athletes provide a unique population (diet, exercise, mental health) prior to injury? They are at the advantage of generally already having good fitness, diet, supplement, mental health and access to trainers etc. Are treatment strategies for sports-concussions being applied in a systemic way anywhere as a case study?
- These questions are now addressed in section 2.4 more explicitly
- Conclusion amended to more specifically address SRCs
- Add a discussion about repeated concussions which is commonly seen in sports-related mTBI cases. Can early prevention strategies help in the case of repeated injuries? If PPCs are not properly diagnosed or left untreated, are consequences exacerbated if an athlete sustains another injury?
- This issue was expanded on in section 2.5.
Specific Comments
38: Can you provide an example? Example provided in the text
44: justify the use of “so-called”; are you referring to limitations of the model? Language revised to avoid the confusion.
51: Lacking a clear definition of central sensitization in chronic pain. Can you expand (take from Section 3.1)? Section 3.1 moved to this area of text and revised.
59: Make a link here to concussions. “We argue that mTBI should also be examined under the conservative care paradigm because…” Sentence added as suggested.
Table 1: Too much information in the summary of findings. Could you add a column for timeline and pull out this information? Adding a column was not feasible due to the heterogenicity of studies, but the information in the summary of findings was simplified.
199-212: Very similar to Section 1.3. Only need to include once. Addressed per above.
229-239: This would be an appropriate place to include more preclinical evidence. Studies in rodents also support these findings (where litigation intent doesn’t matter). Preclinical study reference added.
243-253: This paragraph can be condensed- doesn’t contribute much to your argument. Paragraph was condensed and tie-in to argument made more explicit.
Table 2: Confusing to have two “PS” abbreviations. Can details about timeline be included? Need more of an explanation for which categories are included in the table. Why not include them all? Alternatively, you could restructure the paragraph and list all variables not included in the table at the end. PS error corrected and preceding paragraph amended to clarify the table.
282: define early. Defined.
353: Creatine specifically is a supplement athletes may have already been taking prior to injury. Any evidence about how common this is or if it has any preventative effects if taken prior to concussion. Creatine section expanded on to address your question.
Thanks!
Reviewer 2 Report
Comments and Suggestions for Authors
This is a helpful review of methods in the conservative management of sports related concussions. The authors helpfully frame concussions in the context of pain and provide good explanation of concepts of central sensitization.
There are some small aspects of the manuscript that should be addressed:
Section 2.4 Attempts to explain the effect of age on symptom resolution are not overly clear. In presenting the Toronto Concussion Study are you providing data of adults that took longer to recover than the CARE Consortium data, that was more similar to the 5P study?
Also in this section, the litigation intent is not explained in sufficient detail.
Page 2, Line 79 – PPSS instead of PPCS
Page 4, Line 159 – should read “can be appreciated”
There is no callout in the text referring to Tables 3 through 6. These tables could be removed as they are a simple summary of the preceding information.
There are two tables that are listed as Table 4.
Author Response
Dear Reviewer,
Thank you for your kind remarks and thorough review. Please see below our changes in response to your input.
Sincerely,
John Francis
--------------------------
Section 2.4 Attempts to explain the effect of age on symptom resolution are not overly clear. In presenting the Toronto Concussion Study are you providing data of adults that took longer to recover than the CARE Consortium data, that was more similar to the 5P study? This section was revised to remove the confusion.
Also in this section, the litigation intent is not explained in sufficient detail. Litigation intent explained in more detail.
Page 2, Line 79 – PPSS instead of PPCS. Corrected.
Page 4, Line 159 – should read “can be appreciated”. Corrected.
There is no callout in the text referring to Tables 3 through 6. These tables could be removed as they are a simple summary of the preceding information. There are two tables that are listed as Table 4. Tables 3-6 removed per your request.